# Author Cooperation Network in Biology and Chemistry Literature during 2014–2018: Construction and Structural Characteristics

**Jinsong Zhang [1], Xue Yang [1], Xuan Hu [1] and Taoying Li [1,2,]*** 

[1]  School of Maritime Economics and Management, Dalian Maritime University, Dalian 116026, China
[2]  Logistics Research Institute, Dalian Maritime University, Dalian 116026, China
*   Correspondence: litaoying@dlmu.edu.cn

**Abstract:** How to explore the interaction between an individual researcher and others in scientific research, find out the degree of association among individual researchers, and evaluate the contribution of researchers to the whole according to the mechanism and law of interaction, is of great significance to grasp the overall trend of the field. Scholars mostly use bibliometrics to solve these problems and analyze the citation and cooperation among academic achievements from the dimension of "quantity". However, there is still no mature method for scholars to explore the evolution of knowledge and the relationship between authors; this paper tries to fill this gap. We narrow down the scope of research and focus the research content on the literature in biology and chemistry, collect all the papers from PubMed system (a very comprehensive authoritative database of biomedical papers) during 2014–2018, and take year as a specific analysis unit so as to improve the accuracy of the analysis. Then, we construct the author cooperation networks. Finally, through the above methods and steps, we identify the core authors of each year, analyze the recent cooperative relationships among authors, and predict some changes in the cooperative relationship among the authors based on the networks' analytical data, evaluating and estimating the role that authors play in the overall field. Therefore, we expect that the cooperative authorship networks supported by the complex network theory can better explain the author's cooperative relationship.

**Keywords:** complex network; author cooperation; relationship evolution; knowledge mining

## 1. Introduction

The development of scientific research is never monopolized by a specific country, and it is always created by all the scholars in the world. Scholars differ greatly due to many causes such as countries, research facilities and conditions, research preferences and teams, and so on [1]. The study on the cooperative relationship among authors has been widely used in statistics [2], bibliometrics [3], social sciences [4] and humanities [5]. Focusing on academic papers from a quantitative perspective is an effective way to study the cooperative relationship among authors [6].

Of course, due to the diversity of research methods and analytical perspectives, the results and emphases of different methods are varied. Therefore, choosing the right method for an author cooperation network is very important for effectively reflecting the final result and reasonable explanation. Knowledge graph has shown unparalleled advantages in data analysis since its birth; with its continuous evolution and development, it has gradually evolved into different forms of expression. Researchers found that as an important manifestation of knowledge graph, complex network is a powerful method in the mining and exploration of an authors' cooperative relationship and integrates a large number of discrete things and finds out general rules for scholars. Obviously, the

complex network theory is mostly used in dealing with the cooperative relationship among authors. For example, Rego et al. [7] introduced the link strength theory into the author's network formation model, taking efficiency and stability as the basis for judging the network model. Singh et al. [8] constructed the co-authorship network of Indian physicists to analyze the structure and evolution of the network at 5-year intervals. For the groups to which the co-authors belong, Geraei et al. [9] used social network analysis and small group analysis to investigate the collaboration between different departments and research centers and analyzed and discussed the importance and status of scientific collaboration in medical research. Medina [10] constructed a co-authorship network to assess the collaborative model between ecologists, focusing on the impact of distance and reputational asymmetry on author collaboration. Furthermore, it has been confirmed that there is a relationship among the status of scientists in the collaborative network and their research performance in the fields of pharmacology and nanoscience [11]. Some scholars have taken a different approach. For the research performance of scholars, co-authorship is an important indicator of researcher collaboration skills [12] and is used to study the relationship among journal impact factors [13]. Based on social network analysis, Bellotti [14] studied the relationship between individual and organizational characteristics to reflect the individual's position and value in the team. Andrade, et al. [15] divided the indicators into unweighted, weighted with the weight of the edges, and weighted with weights of the edges and the nodes' attributes to further study the attributes of cooperative networks. Cimenler et al. [16] collected collaborative output data of researchers in a self-reporting way, which can provide some instructions on whether collaborative research is important or not, and improved the authenticity and accuracy of the results to a certain extent. More specifically, Souza et al. [17] used a co-author network to assess the mechanisms of human interaction and productivity performance in specific groups with changes in various network indicators.

The digitization and documentation of scientific papers have enabled the scientific community to establish scientific collaboration and citation networks, and track their proceedings [18]. At present, citation analysis, co-author analysis, co-word analysis, and network analysis of other indicators in the form of knowledge production and scientific discovery are still important methods for bibliometric analysis [19]. Social network analysis has become an important sociological method for discovering network topology attributes [20]. References related to the author cooperation network have provided a reference for the study of academic cooperation. However, there are still some shortcomings in the current cooperative research among authors based on statistical methods and bibliometrics.

Our contributions are to innovatively apply knowledge graph for analyzing the author cooperation relationship, and use the yearly data to track the evolution. Meanwhile, our methods can be easily extended to other fields. We emphasize capturing the evolution of author cooperation in each year to accommodate rapid knowledge updates and focus our attention on the strength of the relationship among authors, then we draw 5 years' author cooperation networks. Finally, we discuss the phenomenon reflected by the networks from five important analytical perspectives.

## 2. Materials and Methods

### 2.1. Data Collection

In this paper, Google scholar [21] was used to find out the top 100 journals cited most in 2018. Then, we employed the PubMed system [22] to query all papers of these journals during 2014–2018, and papers without authors and abstracts were removed. Finally, a total of 77 journals (see in Table 1) and 466,118 papers were obtained because some journals are not included in PubMed system. Subsequently, we extracted authors' names from these papers, which were abbreviations. However, there are too many papers and authors, and it will be very difficult to judge whether there are the same names among different authors, the reasons of which can be presented into two aspects: (1) There may be researchers with the same name even in the same department in the same institute, and only the mailbox can be used to determine whether it is the same person. (2) Many papers only contain the

corresponding author's email; not all papers contain all authors' mailboxes. Therefore, in order to simplify the experiment process, we did not consider the issue of the authors with the same name in this paper.

**Table 1.** Overview of the 77 journals.

| No | Journal Name | IF(5years) | Rank | Area(s) | Press |
|----|--------------|-----------|------|---------|-------|
| 1 | Nature | 44.958 | JCR1 | Multidisciplinary Science | Macmillan Journals ltd. |
| 2 | Chemical Society reviews | 41.27 | JCR1 | Chemistry and Multidisciplinary | Chemical Society. |
| 3 | Cell | 33.796 | JCR1 | Biology | MIT Press. |
| 4 | Nature Communications | 13.691 | JCR2 | Multidisciplinary Science | Nature Pub. Group |
| 5 | Chemical Reviews | 55.198 | JCR1 | Chemistry and Multidisciplinary | American Chemical Society. |
| 6 | Journal of the American Chemical Society | 13.613 | JCR1 | Chemistry and Multidisciplinary | Easton, Pa. [etc.] |
| 7 | Nucleic Acids Research | 10.235 | JCR1 | Biochemistry and Molecular Biology | Information Retrieval ltd. |
| 8 | ACS Nano | 14.82 | JCR1 | Chemistry and Multidisciplinary | American Chemical Society |
| 9 | Physical Review Letters | 7.888 | JCR1 | Physics Multidisciplinary | American Physical Society |
| 10 | Nano Letters | 14.201 | JCR1 | Chemistry and Multidisciplinary | American Chemical Society |
| 11 | Nature Genetics | 31.154 | JCR1 | Genetics and Heredity | Nature Pub. Co. |
| 12 | Journal of the American College of Cardiology | 18.737 | JCR1 | Cardiac and Cardiovascular Systems | Elsevier Biomedical |
| 13 | Plos One | 3.352 | JCR3 | Biology | Public Library of Science |
| 14 | Nature Materials | 47.534 | JCR1 | Chemistry and Multidisciplinary | Nature Pub. |
| 15 | Nature Medicine | 33.409 | JCR1 | Biochemistry and Molecular Biology | Nature Pub. Co. |
| 16 | Circulation | 17.902 | JCR1 | Medical Informatics | American Heart Association |
| 17 | Accounts of Chemical Research | 22.361 | JCR1 | Chemistry and Multidisciplinary | American Chemical Society. |
| 18 | The Astrophysical Journal | 5.402 | JCR1 | Astronomy and Astrophysics | The University of Chicago Press for the American Astronomical Society. |
| 19 | Nature Nanotechnology | 45.815 | JCR1 | Materials Science and Multidisciplinary | Nature Pub. |
| 20 | Nature Biotechnology | 43.271 | JCR1 | Biotechnology and Applied Microbiology | Nature Pub. |
| 21 | Nature Photonics | 38.551 | JCR1 | Physics | Nature Pub. |
| 22 | Nature Methods | 41.934 | JCR1 | Biology | Nature Pub. |
| 23 | BMJ | 2.801 | JCR3 | Medical Informatics | BMJ Publishing Group Ltd |
| 24 | Blood | 12.365 | JCR1 | Medical Informatics | Grune & Stratton [etc.] |
| 25 | The Journal of Materials Chemistry A | 9.531 | JCR1 | Chemistry and Physical | Royal Society of Chemistry Pub. |
| 26 | Scientific Reports | 4.609 | JCR3 | Multidisciplinary Sciences | Nature Publishing Group |
| 27 | Neuron | 16.076 | JCR1 | Medical Informatics | Cell Press |
| 28 | Cochrane Database of Systematic Reviews | 7.669 | JCR2 | Medical Informatics | Oxford, U.K. |
| 29 | Gastroenterology | 19.131 | JCR1 | Medical Informatics | Baltimore. |
| 30 | Nature Neuroscience | 19.188 | JCR1 | Medical Informatics | Nature America Inc. |
| 31 | Advanced Functional Materials | 13.274 | JCR1 | Chemistry and Multidisciplinary | Wiley-VCH, c2001- |
| 32 | Immunity | 23.618 | JCR1 | Medical Informatics | Cell Press |
| 33 | The Journal of Clinical Investigation | 14.434 | JCR1 | Medical Informatics | American Society for Clinical Investigation. |
| 34 | Nanoscale | 7.713 | JCR1 | Chemistry and Multidisciplinary | RSC Pub. |
| 35 | ACS Applied Materials & Interfaces | 8.284 | JCR1 | Nanoscience and Nanotechnology | American Chemical Society |

**Table 1.** *Cont.*

| No | Journal Name | IF(5years) | Rank | Area(s) | Press |
|---|---|---|---|---|---|
| 36 | Monthly Notices of the Royal Astronomical Society | 4.893 | JCR2 | Astronomy and Astrophysics | Oxford University Press |
| 37 | Nature Reviews Immunology | 46.507 | JCR1 | Medical Informatics | Nature Pub. Group |
| 38 | Science Translational Medicine | 18.614 | JCR1 | Cellbiology | American Association for the Advancement of Science Nature Pub. |
| 39 | Nature Reviews Genetics | 44.913 | JCR1 | Genetics and Heredity | Nature Pub. Group |
| 40 | Nature Reviews Cancer | 50.293 | JCR1 | Medical Informatics | Nature Pub. Group |
| 41 | Cell Stem Cell | 23.799 | JCR1 | Cell and Tissue Engineering | Cell Press |
| 42 | Cancer Research | 9.578 | JCR1 | Oncoligy | American Association for Cancer Research |
| 43 | Chemical communications | 6.064 | JCR1 | Chemistry and Multidisciplinary | Royal Society of Chemistry |
| 44 | Nature Climate Change | 22.363 | JCR1 | Environmental Science and Ecology | Nature Pub. Group |
| 45 | Physical Review B | 3.704 | JCR2 | Physics | American Physical Society |
| 46 | Diabetes Care | 10.74 | JCR1 | Biology | American Diabetes Assn. |
| 47 | Advanced Energy Materials | 19.687 | JCR1 | Physics | Wiley-VCH |
| 48 | Hepatology | 11.889 | JCR1 | Medical Informatics | Williams & Wilkins, [c1981]- |
| 49 | Nature Reviews Molecular Cell Biology | 47.918 | JCR1 | Cellbiology | Nature Pub. Group |
| 50 | Annals of Internal Medicine | 18.726 | JCR1 | Medical Informatics | American College of Physicians |
| 51 | Nature Immunology | 21.974 | JCR1 | Medical Informatics | Nature America Inc. |
| 52 | Nature Physics | 22.61 | JCR1 | Physics | Nature Pub. Group |
| 53 | Cell Metabolism | 21.398 | JCR1 | Cellbiology | Cell Press |
| 54 | The Journal of Physical Chemistry Letters | 8.48 | JCR1 | Chemistry and Multidisciplinary | American Chemical Society |
| 55 | The Lancet Neurology | 28.055 | JCR1 | Medical Informatics | Lancet Pub. Group |
| 56 | Environmental Science & Technology | 7.25 | JCR1 | Engineering and Environmental | American Chemical Society |
| 57 | Gut | 15.91 | JCR1 | Medical Informatics | British Medical Assn. |
| 58 | Nature Reviews Neuroscience | 38.691 | JCR1 | Medical Informatics | Nature Pub. Group |
| 59 | European Urology | 15.655 | JCR1 | Medical Informatics | Elsevier Science |
| 60 | Nature Chemistry | 28.79 | JCR1 | Chemistry and Multidisciplinary | Nature Pub. Group |
| 61 | Biomaterials | 9.315 | JCR1 | Engineering and Biomedical | IPC Science and Technology Press |
| 62 | NeuroImage | 7.079 | JCR2 | Medical Informatics | Academic Press |
| 63 | Cancer Cell | 27.072 | JCR1 | Cellbiology | Cell Press |
| 64 | Annals of the Rheumatic Diseases | 11.152 | JCR1 | Medical Informatics | BMJ |
| 65 | Applied Energy | 7.888 | JCR1 | Energy and Fuels | Applied Science Publishers. |
| 66 | IEEE Transactions on Pattern Analysis and Machine Intelligence | 13.229 | JCR1 | Computer Science and Artificial Intelligence | IEEE Computer Society. |
| 67 | Pediatrics | 6.442 | JCR1 | Biology | American Academy of Pediatrics |
| 68 | Journal of Cleaner Production | 6.352 | JCR1 | Environmental Sciences | Butterworth-Heinemann, Ltd |
| 69 | ACS Catalysis | 11.783 | JCR1 | Chemistry and Physical | American Chemical Society |
| 70 | Nature Reviews. Drug Discovery | 54.49 | JCR1 | Biotechnology and Applied Microbiology | Nature Pub. Group |
| 71 | Obstetrical & Gynecological Survey | 2.164 | JCR4 | Medical Informatics | Williams and Wilkins |
| 72 | Circulation Research | 13.313 | JCR1 | Medical Informatics | Lippincott Williams & Wilkins |

**Table 1.** *Cont.*

| No | Journal Name | IF(5years) | Rank | Area(s) | Press |
|---|---|---|---|---|---|
| 73 | Journal of Hepatology | 12.723 | JCR1 | Medical Informatics | Munksgaard International Publishers |
| 74 | The New England Journal of Medicine | 67.513 | JCR1 | Medical Informatics | Massachusetts Medical Society. |
| 75 | JAMA | 10.415 | JCR1 | Medical Informatics | American Medical Association |
| 76 | The Lancet Oncology | 33.234 | JCR1 | Oncology | Lancet Pub. Group |
| 77 | The Astrophysical Journal | 5.402 | JCR1 | Astronomy and Astrophysics | The University of Chicago Press for the American Astronomical Society. |

The author cooperation networks based on these 466,118 papers were constructed and the structural characteristics were analyzed according to the process given in Figure 1.

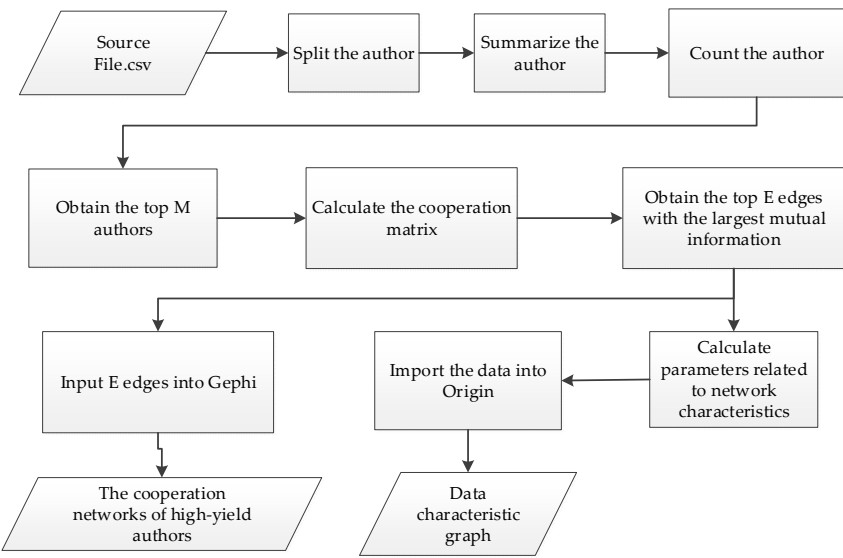

**Figure 1.** The process of constructing and analyzing the author cooperation network.

### 2.2. Author Segmentation and High-Yield Authors

Most of papers have more than three authors, and the total number of authors in these 466,118 papers is close to 1.4 million. We divided the authors of each paper by semicolons and extracted the authors of all papers. All source codes of this study are in the supplementary files. Suppose $N$ is the number of papers in one year, $c_i$ represents the $i$th author, and $n_i$ represents number of papers published by $c_i$ in this year. Then the probability of author $c_i$ in $N$ papers was calculated by $p_i = n_i/N$. We make the authors sort in descending order of $n_i$, ensuring $n_i \geq n_j$ for $\forall i < j$ (equivalent to $p_i \geq p_j$), and the top $M$ authors were selected as high-yield authors. Because of the large number of papers collected, the number of authors is significantly huge. Therefore, in order to make the research results more representative, improve the efficiency of network construction, and simplify the construction process, we ultimately chose high-yield authors to build cooperative networks rather than general authors.

### 2.3. Cooperation Matrix and Cooperation Networks of High-Yield Authors

The cooperative relationship between any two high-yield authors was represented by mutual information in information theory, which describes the degree of cooperation between two authors inspired by Ref. [23]. The mutual information, representing the strength of the relationship between variables [24], was calculated by Equation (1).

$$I_{i,j} = \log_2 \frac{P_{i,j}}{P_i P_j} \tag{1}$$

where $I_{i,j}$ represents the mutual information of authors $c_i$ and $c_j$, $P_{i,j}$ denotes the probability that both of the high-yield authors $c_i$ and $c_j$ are authors of one paper, and $P_i$ and $P_j$ represent the probability of $c_i$ and $c_j$ being authors, respectively. The greater the value of $I_{i,j}$, the greater the cooperation degree between $c_i$ and $c_j$. The matrix $(I_{i,j})_{K \times K}$ is a high-yield author cooperation matrix (considering the symmetric relationship, $I_{i,j} = I_{j,i}$), and it's a diagonal matrix.

In general, we assume that the number of nodes in the cooperation network of high-yield authors is $N$, and the cooperation of authors was expressed as a binary adjacent matrix A ($N$, $N$). If there is a cooperative relationship between two high-yield authors $i$ and $j$, the value of element $a_{ij}$ is 1, otherwise its value is 0. A ($N$, $N$) is a symmetric matrix that can be used to calculate features, such as scale-free effect, small-world feature, hierarchical organization feature, closeness centrality, betweenness centrality, and so on.

### 2.4. The Structural Characteristics of Author Cooperation Networks

#### 2.4.1. Scale-Free Effect

The scale-free network was first proposed by Barabasi and Alber to explain the origin of power law in networks [25]. The degree distribution of complex networks is represented by the probability distribution of node degrees, which offers an effective method for discussing the features of complex networks. The topology and dynamic behavior of complex networks rely on the analysis of their degree distribution [26]. Let $p(k)$ denote the ratio of the number of nodes with degree $k$ to all nodes, then the scale-free effect of the network is expressed by the relationship of $p(k)$ and $k$, satisfying the power-law distribution: $p(k) \sim k^{-\gamma}$. A typical feature of a scale-free network is that only a few core nodes can be connected to a large number of other nodes, and most of the other nodes can only be connected to a small number.

#### 2.4.2. Small-World Feature

In the process of exploring the network model, Watts found that some systems can be highly aggregated like a regular lattice, but have a small feature path length like a random graph. Analogous to the small world phenomenon, he firstly called these systems "small world" networks [27]. The criterion for a small world network is that any two nodes in the network can be reached from each other by a few steps [28]. The small-world feature in a complex network is measured by two indicators, namely the average path length and the clustering coefficient. The average of the shortest distances between all pairs of nodes in the network is called the average path length, where the distance between nodes refers to the minimum number of edges to be connected to these two nodes. The average path length is calculated by Equation (2).

$$L = \frac{\sum_{i \neq j} d_{ij}}{N(N-1)} \tag{2}$$

where $d_{ij}$ represents the shortest distance between nodes $i$ and $j$ in a cooperation network. The aggregation coefficient of a network describes the probability when two neighbor nodes of a node are each other's neighbor nodes, which reflects the partial clustering characteristics of the network. Clustering coefficient is calculated by Equation (3).

$$C = \frac{1}{N} \sum_i \frac{N_i}{k_i(k_i - 1)/2} \tag{3}$$

where $k_i$ is the degree of node $i$, and $N_i$ indicates the number of edges among neighbors of $i$. We assume that the average path length of a Random Network with the same number of nodes and edges to our network is defined as $L_{random}$, while its clustering coefficient is defined as $C_{random}$. If the average path

length and clustering coefficient satisfy the following two conditions, the network exhibits small world feature: $L \approx L_{random}$, and $C >> C_{random}$.

### 2.4.3. Hierarchical Organization Feature

Hierarchical organization is an organizational structure where every entity in the organization, except one, is subordinate to a single other entity [29]. Hierarchical network represents the connectivity among nodes of the real world network. The change of the average degree $k$ and its corresponding clustering coefficient $C(k)$ follow the power-law distribution: $C(k) \sim k^{-\theta}$, where $\theta > 0$, which is a condition in which the network has a hierarchical structure. This formula describes the fact that if the degree of some nodes is lower and the aggregation coefficient is higher, they are high-connected modules. However, some nodes belong to low-connected modules even if they have a higher degree and lower aggregation coefficient. In a hierarchical organization network, some nodes in small scale are loosely connected to form larger modules.

### 2.4.4. Closeness Centrality

The concept of closeness centrality was first proposed by American sociologist Freeman who put forward closeness as a measure of global centrality in terms of the distance among various nodes. The closeness centrality reflects the center extent of a node and its indirect influence on other nodes, and it is expressed as the reciprocal of the cumulative shortest path from one node to other nodes [30]. When information begins to spread from the central nodes, it will be transmitted from the network center to other corners at a fast speed [31]. A node has a high closeness centrality, which means that it is located at the center of the network and is closer to other nodes. The Equation (4) for calculating the closeness centrality of node $i$ is as follows.

$$C_C(i) = \frac{N}{\sum_{j \in N} d_{ij}} \tag{4}$$

### 2.4.5. Betweenness Centrality

The betweenness centrality of node $v$ is calculated by Equation (5):

$$C_B(v) = \sum_{i \neq v \neq j \in N} \frac{d_{ij}(v)}{d_{ij}} \tag{5}$$

where $d_{ij}(v)$ represents the number of paths through node $v$ in $d_{ij}$. It is another concept of node centrality proposed by Freeman, which measures the extent to which a node is located in the middle of other "node pairs" in the network [30]. To be more precise, for a given node, it measures how many of the shortest paths pass through it [32]. The betweenness centrality reflects the importance of a node to information transfer. A node with a high betweenness centrality means that it acts as an indispensable "mediator" in the process of information dissemination.

## 3. Results and Discussion

### 3.1. Cooperation Network of High-Yield Authors in Biology and Chemistry

In descending order of cooperation matrix and mutual information, the threshold of cooperation between high-yield authors was set to $E$, and then the top $E$ cooperation became the number of edges of the network. The authors corresponding to these $E$ edges act as nodes, the number of these authors is represented by the letter $M$, and the network composed of these authors and edges is the cooperation network of high-yield authors. In this paper, the authors of 466,118 papers were used to extract the top 1000 high-yield authors each year and construct networks, which mean that the value of $M$ was 1000. In addition, we chose 800 as the value of $E$. Through comparative experiments, it was found that when

*M* was fixed, increasing *E* had a little effect on network characteristics; when *E* was fixed and *M* was increased, the network remained unchanged. When the values of *M* and *E* increased simultaneously, compared with the network of *E* = 800 and *M* = 1000, the visualization was poor due to too many nodes and edges. Simultaneously, added nodes and edges increased the complexity of the network making some structural features not clearly reflected. As a result, the top 1000 high-yield authors were selected. Among these 1000 high-yield authors, the largest 800 mutual information were selected as the edges. The nodes and edges obtained above were used to build five cooperation networks by year. The final network graphs are shown in Figure 2.

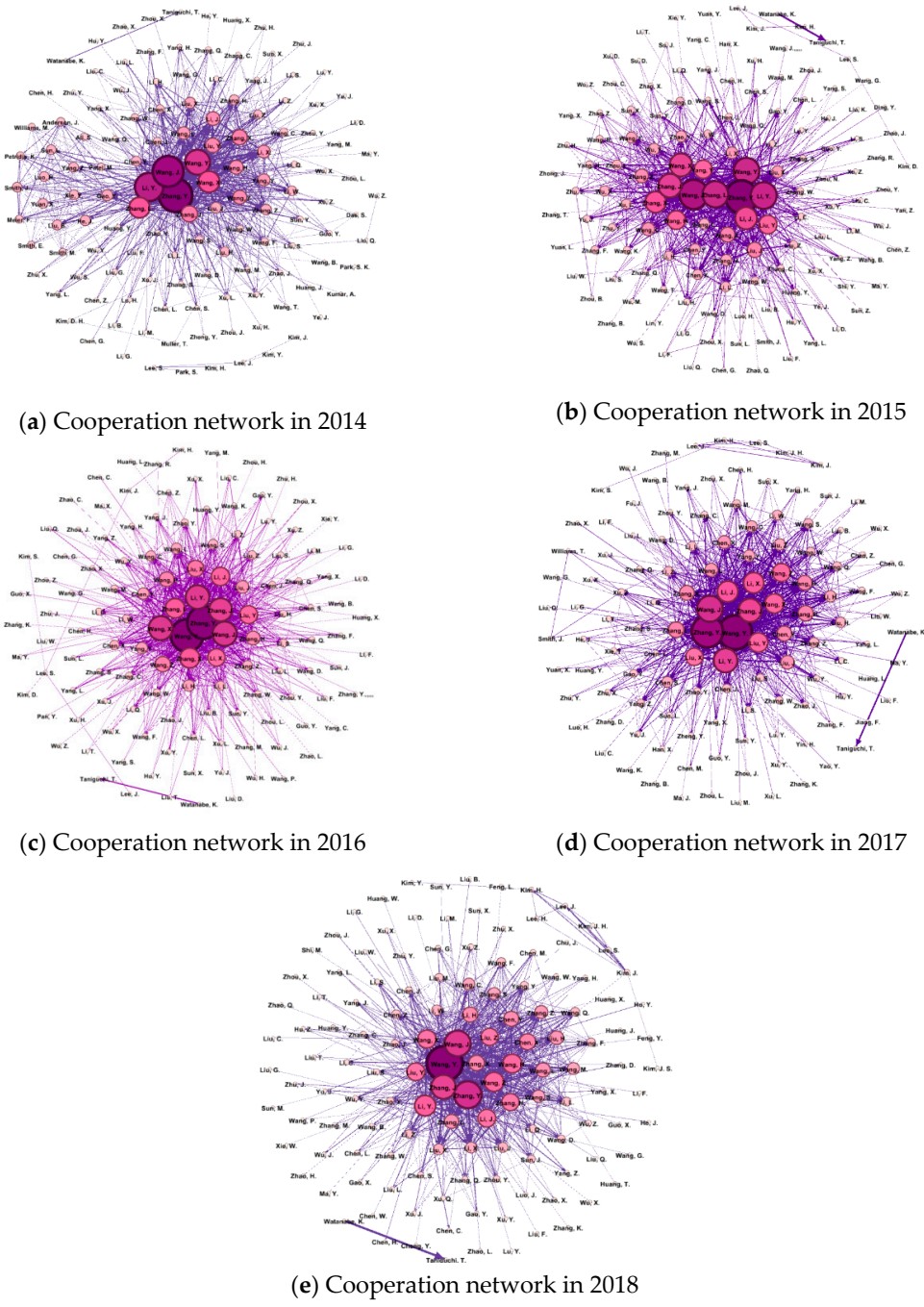

**Figure 2.** Cooperation network graphs among authors during 2014–2018. Figure 2 contains five graphs, which are arranged by year as: (**a**) 2014, (**b**) 2015, (**c**) 2016, (**d**) 2017, (**e**) 2018.

In Figure 2, the value of *M* is 1000 and the threshold *E* is 800; the number of nodes in the network ranges from 120 to 140, which is relatively stable. Network density can be used to characterize the degree of interconnection between nodes, defined as the ratio of the number of edges actually present in the network to the upper limit of the number of edges that can be accommodated. The density of the network in each year is 0.092, 0.085, 0.098, 0.105, and 0.091, respectively, which means that the values of the extent of potential relationship realization in the cooperation network from 2014 to 2018 are 9.2%, 8.5%, 9.8%, 10.5%, and 9.1%, respectively. The range of variation is around 1%. First, the interaction of the research team is the antecedent of the author's cooperation network. Active participation in scientific collaboration can affect the density of the network relationship structure. Second, when the density is higher, the more connections, information and human resources the author can get, and vice versa. Thirdly, retirement/appointment, enrollment/graduation, research team turnover, and internal staff mobilization are the main factors affecting the density of the network structure, which further influences interactive behavior and knowledge dissemination. All in all, the withdrawal of some people is always accompanied by the addition of others, keeping the network density in a dynamic and stable state.

### 3.2. Scale-Free Effect

The distribution of node degrees of the graphs during 2014–2018 and the change of statistical indicators of degree centrality are shown in Figures 3 and 4 respectively. In Figure 3, the abscissa in the coordinate axis of the graph represents the degree of the node, and the ordinate represents the proportion of nodes having degrees greater than or equal to *k* to the number of total nodes. It is obvious that the cumulative degree distribution from 2014 to 2018 is not a straight line but a curved curve that conformed to the power-law distribution. Therefore, the cooperation networks from 2014 to 2018 are all scale-free networks.

In a complex network, the phenomenon that the degree is divided into two different segments implies the existence of a nuclear dictionary [33]. Despite the small portion of nodes, they are connected to a large number of other nodes, and most of the remaining nodes are connected to only a few nodes. Similarly, in the cooperation network, the curve is divided into two stages, which means that there are core authors in the biological and chemical papers. The advancement of scientific study relies to a large extent on individual scholars who have made tremendous contributions to the discipline by conducting their research teams. Although they are few in number, they have extensive cooperation with other authors. In Figure 3, the annual curve has not changed significantly compared with the previous year, which indicates that the scale-free effect of the cooperation network is stable year by year. We tried to use the scale-free structure to screen out the leading core authors. Table 2 contains the core authors of individual year durign 2014–2018. For instance, the top three authors in 2014, the author "Zhang, Y.", who ranked first in the ranking of core authors in 2014–2016, slipped to second in 2017. Author "Wang, J.", ranked second before 2015 and third in 2016. Similarly, author "Li, Y." has floated between third and fourth place since 2016. Other authors, such as author "Wang, Y.", have taken the top spot since 2017. Due to the large number of authors in this paper, it is unavoidable that there are authors with the same abbreviated names. In this case, we use the authors' names and the affiliations in Table 2 as the search conditions, and calculate the number of papers published by authors under the same abbreviated names shown in Table 3. Although there are authors with the same name in Table 3, it can be seen from the number of published papers that each name has an author who clearly publishes more papers than others, so the conclusions studied in this paper are still valid. It is easy to find that the number of papers by a few authors is significantly higher than that by other authors. Therefore, we believe that the high-yield authors in this paper come from those who have a significantly higher volume of publications. Figure 4 shows the degree centrality of the nodes in the network. We can observe that since 2015, the centrality has shown a slow upward trend. The greater the degree centrality, the more nodes in the network that have direct contact with other nodes, and the higher the author participation in the network. Also, it proves that the core authors

in the network are more and more concentrated, and the number of connections between them far exceeds that of connections between other nodes. In addition, according to the changes of the three lines given in Figure 4, although the mean of degree centrality has increased slightly over the years, the median of degree centrality has remained stable, which shows that the distribution of degree centrality has become more and more asymmetric over the years. Table 4 gives the exponential change of the power-law distribution shown in Figure 3. The value of exponent $\gamma$ has a slight downward trend, meaning the curve decreases in amplitude. In the case of increased network complexity, the increase in nodes with higher degrees is slightly larger than the increase in nodes with low degrees.

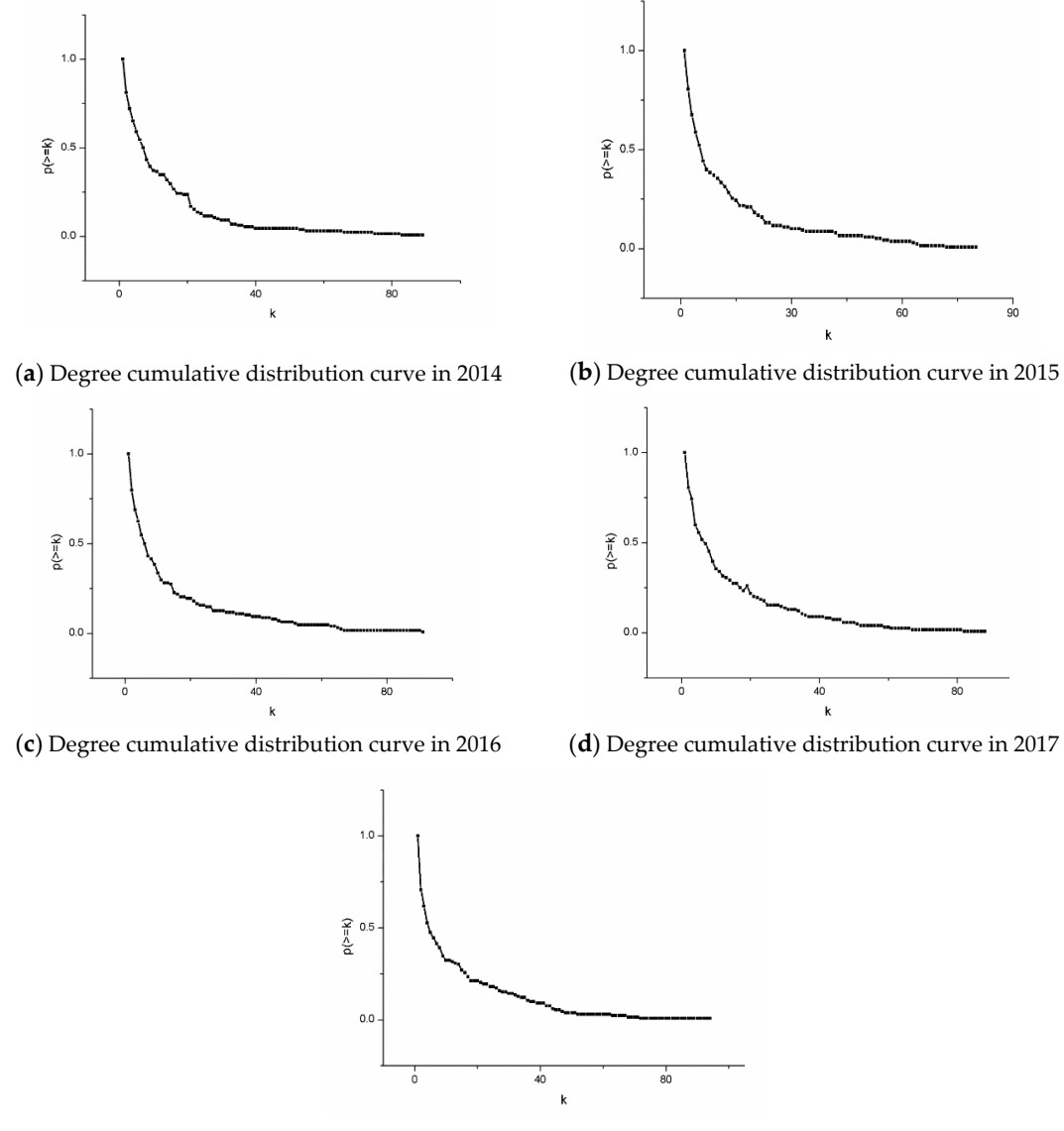

(**a**) Degree cumulative distribution curve in 2014     (**b**) Degree cumulative distribution curve in 2015

(**c**) Degree cumulative distribution curve in 2016     (**d**) Degree cumulative distribution curve in 2017

(**e**) Degree cumulative distribution curve in 2018

**Figure 3.** Degree cumulative distribution curve during 2014–2018. Figure 3 contains five graphs, which are arranged by year as: (**a**) 2014, (**b**) 2015, (**c**) 2016, (**d**) 2017, (**e**) 2018.

Without changing the network density, the author cooperation network is composed of a small number of core authors and a large number of non-core authors. The cooperation relationship extended by the core authors constitutes the framework of the whole network. Knowledge exchange in the network is mostly accomplished through the transformation between the main paths in the framework. Meanwhile, among the biology and chemistry papers, the core authors have not changed much in recent years and are likely to remain stable in the next few years. Only the ranking order of core

authors changed. Since some authors are unable to publish papers because of many reasons, some change the research areas owing to the extension and shift of the research direction. At the same time, authors who have a relatively stable research work in the field and a close group connection have also increased the number of published papers. These have resulted in changes in the ranking order of core authors. However, due to the improvement of the education level of various countries, the emphasis on knowledge and the treatment of scientific research talents have been strengthened. When some high-level and creative talents emerge in the field, such a situation may be broken. In addition, with the general increase of degree centrality of nodes, the complexity of a whole network is strengthened, leading the cooperation between authors to increase. This is an inevitable trend of vigorous development in the field of biology and chemistry.

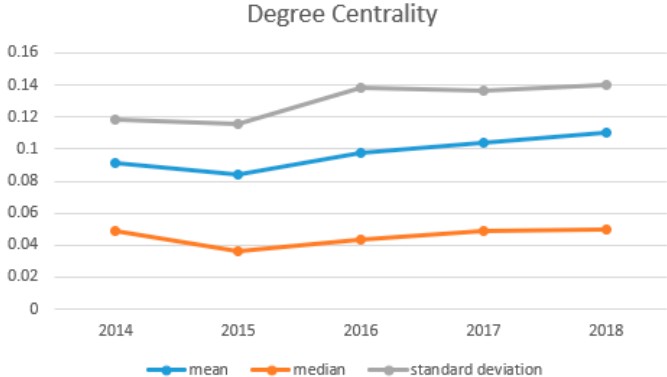

**Figure 4.** The statistical indicators of degree centrality curve during 2014–2018.

**Table 2.** The core authors of each year.

| 2014 | 2015 | 2016 | 2017 | 2018 |
|------|------|------|------|------|
| Zhang, Y. | Zhang, Y. | Zhang, Y. | Wang, Y. | Wang, Y. |
| Wang, J. | Wang, J. | Wang, Y. | Zhang, Y. | Zhang, Y. |
| Li, Y. | Li, Y. | Wang, J. | Wang, J. | Wang, J. |
| Wang, Y. | Wang, Y. | Wang, X. | Li, Y. | Zhang, J. |
| Wang, X. | Zhang, L. | Li, Y. | Zhang, J. | Li, Y. |
| Zhang, L. | Zhang, J. | Zhang, J. | Li, J. | Wang, Z. |
| Zhang, J. | Wang, X. | Zhang, X. | Li, X. | Wang, H. |
| Zhang, X. | Li, J. | Zhang, L. | Liu, Y. | Wang, X. |
| Liu, Y. | Zhang, X. | Li, J. | Wang, X. | Li, J. |
| Li, J. | Liu, Y. | Liu, Y. | Wang, Z. | Liu, Y. |

**Table 3.** Statistical data on the number of papers published by authors from different affiliations with the same abbreviated names.

| Author | Number of Papers | Affiliation |
|--------|------------------|-------------|
| Zhang, Y. | 3852 | University of Chinese Academy of Science |
| | 796 | QiLu Hospital |
| | 392 | The Third Affiliated Hospital of Sun Yat-sen University |
| | 353 | National University of Defense Technology |
| | 326 | Steven Institute of Technology |
| Wang, J. | 3245 | Capital Medical University |
| | 691 | The Chinese University of Hong Kong |
| | 447 | University of Illinois at Urbana-Champaign |
| | 343 | Sichuan University |
| | 231 | South China University of Technology |

**Table 3.** *Cont.*

| Author | Number of Papers | Affiliation |
|---|---|---|
| Li, Y. | 2949 | Shandong University |
| | 870 | Harbin Institute of Technology |
| | 681 | University of Manchester |
| | 657 | Hubei University of Medicine |
| | 453 | Chinese PLA General Hospital |
| Wang, Y. | 3119 | Fudan University |
| | 687 | University of Chinese Academy of Sciences |
| | 537 | Ningbo University |
| | 457 | Cornell University |
| | 439 | Tsinghua University |
| Wang, X. | 3333 | Zhejiang University |
| | 789 | Chinese Academy of Agricultural Sciences |
| | 654 | University of Michigan |
| | 491 | China Medical University |
| | 452 | Xi'an Jiaotong University |
| Zhang, L. | 3619 | Second Military Medical University |
| | 866 | Qingdao University |
| | 762 | Soochow University |
| | 459 | University of Illinois |
| | 433 | Aarhus University |
| Zhang, J. | 3543 | Chinese Academy of Sciences |
| | 784 | Sichuan University |
| | 598 | Shanghai Jiaotong University |
| | 463 | China Agricultural University |
| | 456 | Southeast University |
| Zhang, X. | 3329 | Nankai University |
| | 866 | University of Science and Technology of China |
| | 713 | Harbin Institute of Technology |
| | 772 | Tongji University School of Medicine |
| | 323 | Sichuan Agricultural University |
| | 218 | South China University of Technology |
| Liu, Y. | 2854 | Peking University |
| | 589 | University of Maryland |
| | 545 | East China University of Science and Technology |
| | 371 | Northeast Agricultural University |
| | 292 | The University of Texas at Austin |
| Li, J. | 3074 | University of California |
| | 609 | Jinan University |
| | 580 | Tianjin University |
| | 501 | Lanzhou University |
| | 339 | University of Washington |
| Li, X. | 2468 | Fudan University |
| | 576 | Lanzhou University |
| | 483 | Jiangnan University |
| | 476 | Shanghai Medical College |
| | 273 | Southwest University |
| Wang, Z. | 2778 | China Medical University |
| | 647 | Duke University |
| | 589 | University of Oklahoma |
| | 542 | Nanjing Agricultural University |
| | 449 | Nankai University |
| Wang, H. | 2672 | Tsinghua University |
| | 538 | Massachusetts General Hospital |
| | 393 | Northwestern University |
| | 255 | South China University of Technology |
| | 240 | Xi'an Jiaotong University |

**Table 4.** $\gamma$ for each year.

| Exponent | 2014 | 2015 | 2016 | 2017 | 2018 |
|:---:|:---:|:---:|:---:|:---:|:---:|
| $\gamma$ | 0.592 | 0.601 | 0.593 | 0.575 | 0.568 |

*3.3. Small-World Feature*

For the cooperation networks of high-yield authors in 2014–2018, the average path length and clustering coefficient are shown in Table 5. If the average path length and clustering coefficient satisfy the following conditions simultaneously, it is a small-world network: $L \approx L_{random}$, and $C >> C_{random}$.

**Table 5.** The average path length and aggregation coefficient of the networks during 2014–2018.

| Parameter | 2014 | 2015 | 2016 | 2017 | 2018 |
|:---:|:---:|:---:|:---:|:---:|:---:|
| $L$ | 2.140 | 2.143 | 2.007 | 2.031 | 2.257 |
| $C$ | 0.631 | 0.638 | 0.628 | 0.645 | 0.624 |

In Table 5, the average path length in each year is about 2.1, and the clustering coefficient is between 0.6 and 0.7. Compared with ER random network, the above two conditions are satisfied. Therefore, the cooperation networks from 2014 to 2018 are typical small-world networks, and all of them almost remain at the same level. The average path length reflects the distance and efficiency of knowledge transfer between authors in the network. Small-world network with an average path length of close to 2 confirms that if there is a direct or indirect cooperative relationship between any two authors, there are at most two other authors in the path of connection, which ensures that nodes can be connected to each other within short paths, and is of great significance for knowledge transfer and dissemination. In the case where the clustering coefficient and the average path length are both stable, the small world feature of the network illustrates the cooperation between authors as a means of information transmission in the context of biological and chemical research, and information spreads between authors at a steady rate. When an author has the conditions to collaborate with other authors, his approach to the latest theory in the field is to directly or indirectly contact two or three peer authors. It provides an opportunity to grasp whether individual authors share the same research orientation. Furthermore, it offers a reference for discovering the author's group and core figures in the field. For example, if you want to search the information about author "Wang, Y.", the search system can directly recommend "Zhang, Y." and "Wang, J." because these three authors not only have a large number of papers published from 2014 to 2018, but also have direct cooperative relationship in many papers; they probably belong to one research group.

*3.4. Hierarchical Organization Feature*

Let $C(k)$ denote the average aggregation coefficient of nodes with degree $k$. If $C(k) \sim k^{-\theta}$ and $\theta > 0$, the network has a hierarchical organization [34]. Figure 5 shows the aggregation coefficient of the cooperation network for each year. Table 6 lists the $\theta$ of each year. As can be seen from the figure, in 2014–2015, the $\theta$ of the network is greater than 0, and decreases gradually along the year. Therefore, the average aggregation coefficient distribution of above five cooperation networks conforms to the power-law distribution, and the networks present a hierarchical organization characteristic. The distribution of the aggregation coefficient shows a downward trend, which means that there are not only nodes with a low degree and high aggregation coefficient in the cooperation network, but also nodes with a high degree and low aggregation coefficient. Simultaneously, $\theta$ decreases year by year, which means that more and more nodes with low degree are connected to high-connection nodes making the scale of high-connection module larger.

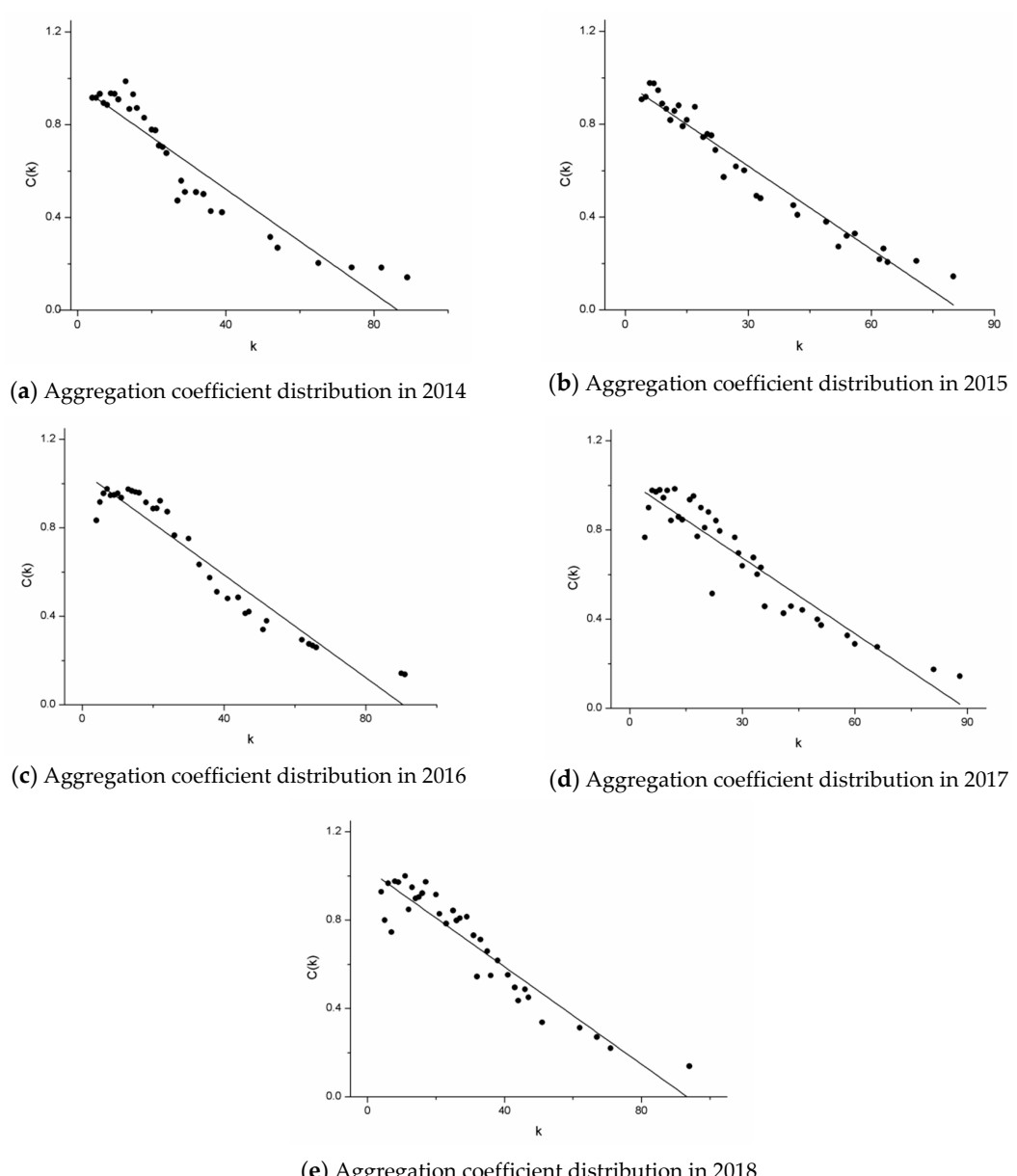

**Figure 5.** Aggregation coefficient distribution during 2014–2018. Figure 5 contains five graphs, which are arranged by year as: (**a**) 2014, (**b**) 2015, (**c**) 2016, (**d**) 2017, (**e**) 2018.

**Table 6.** $\theta$ for each year.

| Parameter | 2014 | 2015 | 2016 | 2017 | 2018 |
|---|---|---|---|---|---|
| $\theta$ | 0.616 | 0.594 | 0.569 | 0.521 | 0.472 |

　　In the cooperation network of high-yield authors, nodes with higher connectivity constitute high-connection modules, while nodes with lower connectivity constitute low-connection modules. We can infer that when some high-yield authors belong to the same group, they often possess consistent research directions and creative content, so they are more likely to have a partnership. Furthermore, the exchange of ideas between some large-scale research groups directly improves the connectivity of the network, and these authors constitute some higher connectivity modules. A part of small-scale research groups or individuals interact and cooperate to some extent constituting low-connected modules. In addition, it is worth noting that the number of authors constituting the high-connection module is steadily expanding. The connectivity module reflects the distribution and interconnection of small

networks that are clustered in a large network. We believe that due to the influence of diversification and complication on some high-yield authors, part of research groups work together to form a larger high-linking module to promote the development of biology and chemistry in a more stable direction.

### 3.5. Closeness Centrality

For a node in the network, its closeness centrality varies from 0 to 1. The node is far away from other nodes while the closeness centrality approaches 0. Conversely, when the closeness centrality approaches 1, the node is close to other nodes [35]. The relationship between the degree of the node and closeness centrality is shown in Figure 6. The phenomenon embodied in the figure is that the closeness centrality of most nodes is positively correlated with the degree. With 0.6 as the demarcation, nodes with closeness centrality higher than 0.6 are closer to all nodes, and such nodes occupy a small portion of the cooperation network. Figure 7 shows how closeness-centralized statistical indicators change over the years. It is not difficult to see that since 2015, the indicators have experienced a brief rise and finally shown a stable state.

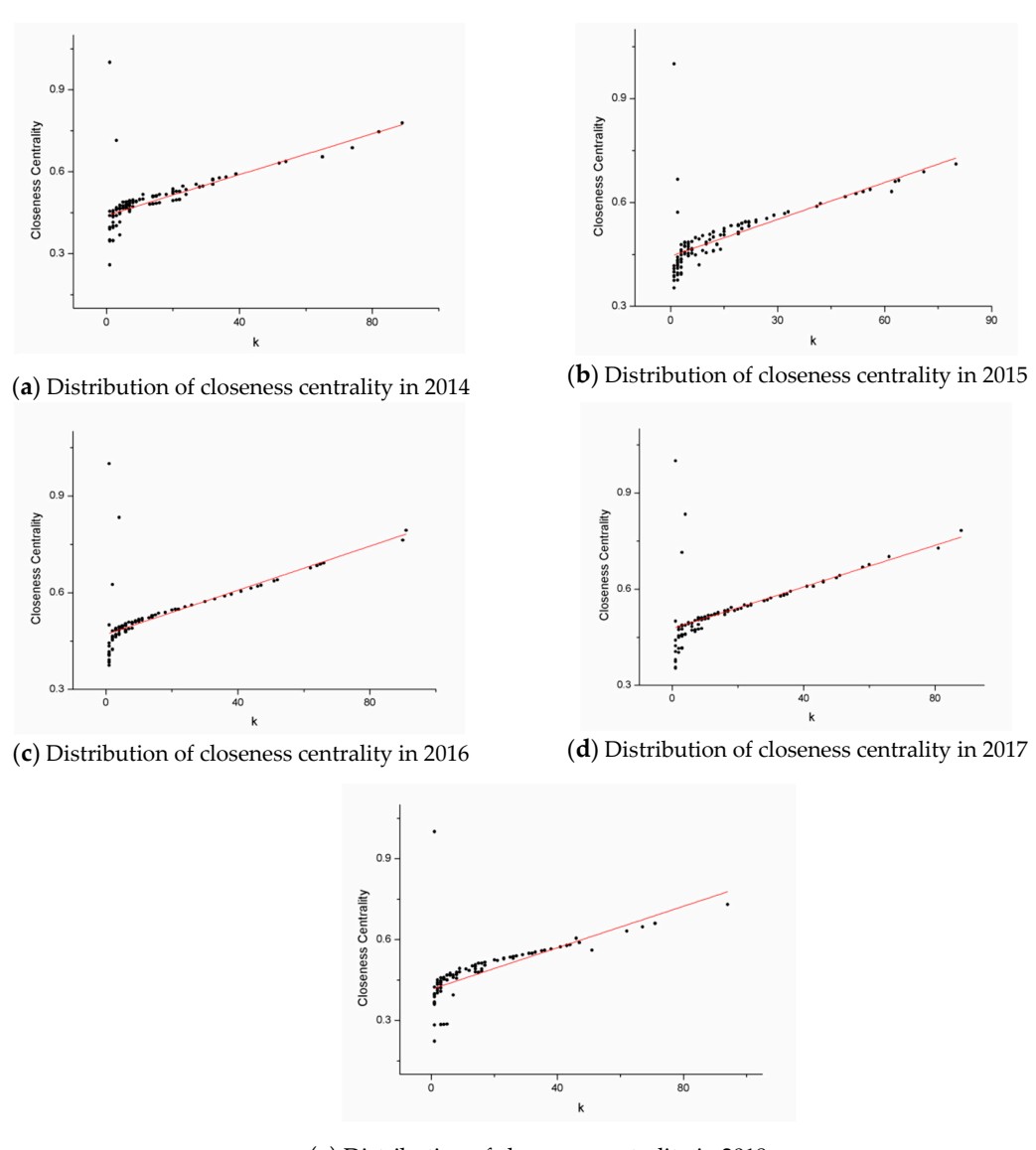

(**a**) Distribution of closeness centrality in 2014

(**b**) Distribution of closeness centrality in 2015

(**c**) Distribution of closeness centrality in 2016

(**d**) Distribution of closeness centrality in 2017

(**e**) Distribution of closeness centrality in 2018

**Figure 6.** Distribution of closeness centrality during 2014–2018. Figure 5 contains five graphs, which are arranged by year as: (**a**) 2014, (**b**) 2015, (**c**) 2016, (**d**) 2017, (**e**) 2018.

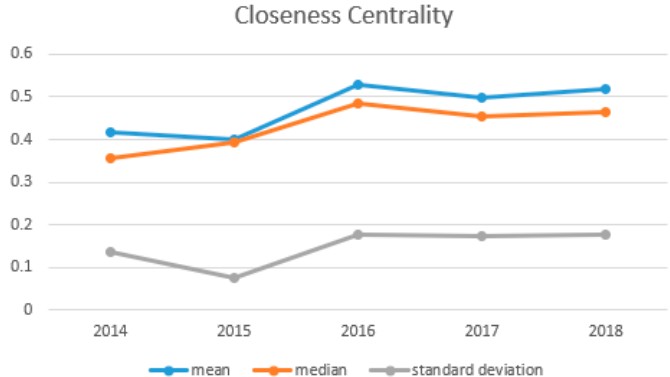

**Figure 7.** The statistical indicators of closeness centrality curve during 2014–2018.

The shortest distance from one node to the other decreases as its closeness centrality increases. Such a node is near the center of the network. The central nodes can quickly transmit information to other nodes in the network. There are more than half of the high-yield authors on the edge of the cooperation network, while a small number of authors occupy the center and have a relatively short distance from other nodes. In the biology and chemistry papers, some of the latest innovations and discoveries are proposed and published by high-yield authors at the network center. Through direct or indirect communication and cooperation with other high-yield authors, the new theories and achievements can be rapidly disseminated. From the change of the data in Figure 6, this situation remains stable within a certain range, indicating that in the most recent 5 years, the authors of the network center have indeed promoted the dissemination of the latest research results in the field of biology and chemistry. Similarly, the data in Figure 7 also confirm the stable state of the networks' closeness centrality, and the authors in the network center will not change significantly in a short time.

*3.6. Betweenness Centrality*

Figure 8 shows the relationship between betweenness centrality and node degree. The betweenness centrality of each network is less than 0.3. From the perspective of quantitative changes, there is no significant difference in the trend of each year except for a few nodes. Nodes with a large degree usually have a higher level of betweenness centrality. Betweenness centrality of nodes with degree less than 30 is almost zero, while that of nodes with degree higher than 30 increases with the increase of node degree. Figure 9 shows the change in the betweenness centrality more intuitively.

Since biology and chemistry are two disciplines with a large extent of knowledge overlap and interoperability, their research subjects will involve many sub-disciplines such as cell biology, medical informatics, biochemistry, and molecular biology, etc. As a result, there must be intersection between these branches. Some nodes with high degree in the cooperation network have strong intermediation, and the high-yield authors corresponding to these nodes are on the shortest path in which some other authors cooperate. These authors play a role in connecting sub-disciplines throughout the cooperation network, and they have extensive communication with authors of different branches and publish papers with them. Figure 9 shows the change of the betweenness centrality more intuitively. From the perspective of various indicators, since 2015, the betweenness centrality has always been in the same state, which shows that among the authors, there are people who always play the role of bridges between others, ensuring that most branches of the field in biology and chemistry are able to communicate with each other and coordinating development, promoting the diversity and vitality of the advance in biology and chemistry

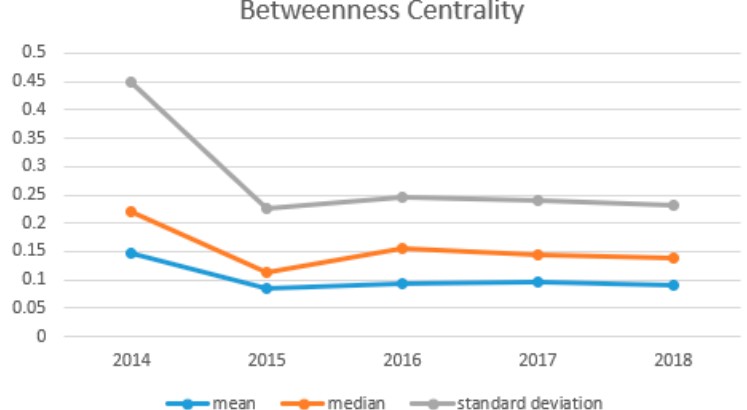

(**a**) Distribution of betweenness centrality in 2014

(**b**) Distribution of betweenness centrality in 2015

(**c**) Distribution of betweenness centrality in 2016

(**d**) Distribution of betweenness centrality in 2017

(**e**) Distribution of betweenness centrality in 2018

**Figure 8.** Distribution of betweenness centrality during 2014–2018. Figure 8 contains five graphs, which are arranged by year as: (**a**) 2014, (**b**) 2015, (**c**) 2016, (**d**) 2017, (**e**) 2018.

**Figure 9.** The statistical indicators of betweenness centrality curve during 2014–2018.

## 4. Conclusions

Investigation and exploration of cooperation relationships in the current field can help scientific decision makers to determine research priorities, improve the structure of human and material resources, and further enhance their contributions to advanced theories, experiments and applications. In order to show the cooperation accurately, we analyzed the change of each year's characteristics and discussed the reasons by using the relevant methods of knowledge graphs. The data we got has been counted and were mapped into networks. The network structure from five perspectives, including scale-free effect, small-world feature, hierarchical organization characteristic, closeness centrality, and betweenness centrality is evaluated, and the conclusions are drawn as following:

- The cooperation density of the network has not changed significantly by year, and it is in a state of dynamic balance.
- The cooperation network is a small-world network with scale-free effect. There are a small number of core authors in the network.
- The direct or indirect cooperative relationship between any two authors goes through at most two other authors.
- Authors in large-scale research groups in the network connect with each other to form high connectivity modules, while some relatively smaller groups or authors form low connectivity modules.
- More than half of the authors are on the edge of the network; by spreading from the center of the network, the latest theories and achievements can be quickly passed to other authors.
- There are always some authors who act as intermediaries, linking various branches of biology and chemistry in the network.

However, we did not consider the question of the same name among different scholars in this paper. In future work, we will consider adding the mailbox for judging if the authors with the same name are the same person.

**Supplementary Materials:** The following are available online at http://www.mdpi.com/2078-2489/10/7/236/s1.

**Author Contributions:** Conceptualization: J.Z. and X.Y.; methodology: T.L.; validation: X.Y. and J.Z.; formal analysis: X.Y.; investigation: J.Z.; resources: J.Z.; data curation: T.L.; writing—original draft preparation: X.Y.; writing—review and editing: J.Z. and X.H.; visualization: X.Y.; supervision: T.L.; project administration: J.Z.; funding acquisition: J.Z.

**Funding:** This research was funded by the National Natural Science Foundation of China, grant number 71271034, the National Social Science Foundation of China, grant number 15CGL031, the Fundamental Research Funds for the Central Universities, grant numbers 3132019028, 3132019175 and 3132019233, the Program for Dalian High Level Talent Innovation Support, grant number 2015R063, the National Natural Science Foundation of Liaoning Province, grant number 20180550307, the China Postdoctoral Science Foundation, grant number 2016M591421, and the National Scholarship Fund of China for Studying Abroad.

**Acknowledgments:** We thank the editor and reviewers for their thorough reviews, thoughtful comments and constructive suggestions.

**Conflicts of Interest:** The authors declare no conflict of interest.

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
