# Peer review of "Author Cooperation Network in Biology and Chemistry Literature during 2014–2018: Construction and Structural Characteristics"

_information, doi:10.3390/info10070236_

Round 1

Reviewer 1 Report

Page 2, lines 46-53. There are other papers regarding the study of co-authorship network which should be included in the reference list. Below I give a fez examples:

1) SOUZA, F. C. ; AMORIM, R. M. ; RÊGO, L.C.. A Co-authorship network analysis of CNPq?s productivity research fellows in the probability and statistic area. Perspectivas em Ciência da Informação (on line), v. 21, p. 29-47, 2016.

2) ABBASI, A.; ALTMANN, J.; HOSSAIN, L. Identifying the effects of co-authorship networks on the performance of scholars: a correlation and regression analysis of performance measures and social network analysis measures. Journal of Informetrics, v. 5, n. 4, p. 594-607, 2011. [ Links ]

3) BELLOTTI, E. Getting funded. multi-level network of physicists in Italy. Social Networks, v. 34, n. 2, p. 215-229, 2012. [ Links ]

4) BORDONS, M. et al. The relationship between the research performance of scientists and their position in co-authorship networks in three fields. Journal of Informetrics, v. 9, n. 1, p. 135-144, 2015. [ Links ]

5) BALES, M. E. et al. Associating co-authorship patterns with publications in high-impact journals. Journal of Biomedical Informatics, v. 52, p. 311-318, 2014. [ Links ]

6) CIMENLER, O.; REEVES, K. A.; SKVORETZ, J. A regression analysis of researchers’ social network metrics on their citation performance in a college of engineering. Journal of Informetrics, v. 8, n. 3, p. 667-682, 2014. [ Links ]

7) DIGIAMPIETRI, L.; REGO, LEANDRO, C.; SOUZA, F. C. ; OSPINA, R. ; MENA-CHALCO, J. . Brazilian network of PhDs working with probability and statistics. Brazilian Journal of Probability and Statistics, v. 32, p. 755-782, 2018.

8) ANDRADE, R. L. ; REGO, LEANDRO C. . EXPLORING THE CO-AUTHORSHIP NETWORK AMONG CNPQ?S PRODUCTIVITY FELLOWS IN THE AREA OF INDUSTRIAL ENGINEERING. PESQUISA OPERACIONAL (IMPRESSO), v. 37, p. 277-310, 2017.

More generally, there is the following review paper:

9) KUMAR, S. Co-authorship networks: a review of the literature. Aslib Journal of Information Management, v. 67, n. 1, p. 55-73, 2015. [ Links ]

Page 6, line 112. The letter K was already used in line 86 with another meaning. Please use a different letter to avoid confusion.

Page 7, lines 129-130. It is unclear what the authors meant by random probability in this phrase. Please, give more details.

Page 7, lines 146-147 and lines 153-154. Please, revise English language of these phrases.

Page 9, line 192. Replace "cumulative distribution" by "distribution of node degrees of the".

Page 9, 194. Replace "ration of the number of nodes corresponding to the degree in all nodes" by "proportion of nodes having at least the degree in the abscissa".

Page 9, lines 225-226. Revise English.

Page 9, it would be interesting to have information about the evolution of the mean, median and standard deviation of the  degree centralities of the nodes of the neworks along the years.

Page 10, it would be good to report the exponent, gamma, of the power-law distribution for each one of the years to see its evolution.

Page 11, lines 260-261. Although all of the theta values are greater than 0, none of them seems to be close to 1, as stated. On the other hand, it should be mentioned that \theta is steadly decreasing in the analyzed period.

Page 13, it would be interesting to have information about the evolution of the mean, median and standard deviation of the closeness centralities of the nodes of the neworks along the years.

Page 14, it would be interesting to have information about the evolution of the mean, median and standard deviation of the betweeness centralities of the nodes of the neworks along the years.

Page 15, lines 346-347. Not distinguishing the same name among different people is a serious limitation that might put in check the validity of the results presented. Therefore, I suggest that authors should at least take care to investigate whether the same name appears in the case of the reserchers shown in Table 2, which are the most relevant for the analysis. 

Author Response

We have uploaded a point-by-point response to the reviewer’s comments.

Reviewer 2 Report

The paper "Author Cooperation Network in Biology and Chemistry Literature during 2014-2018: Construction and Structural Characteristics" presents and interprets a few network metrics on five small coauthorship constructed for the years 2014-2018.

My comments and suggestions are as follows:

section 2.1:

- you should state here that you do not consider the case of persons with the same name, you should do this as the presented conclusions may be invalid. Also, did you check if someone changes her name?

- why use google scholar and PubMed and not Web of Science or Scopus databases?

- you use google scholar to find the top 100 cited journals in 2018 but limit your search to 77? What was your criteria for eliminating 23 journals?

- why do you consider only the year 2018 when searching for the top 100 most cited journals? does this list remain unchanged during the period 2014-2018? if not, what would change if you construct the networks based on the top cited journals for each considered year?

- if the above list changes table 1 brings no relevant information to the reader

- please be consistent in your notations, eg. in figure 1 N represents the top number of authors considered, in section 2.2 K is used for the top number of authors and N for the number of papers considered

page 5, section 2.2

- I think your pseudocode is actually python code, please present it in an algorithmic way, also give appropriate names (see A4). If you want to reference your code (and you should do this for reproducibility purposes) you should post your code on a web-based hosting service like github, bitbucket, gitlab and give a link to your code. Also consider posting the obtained networks.

- please rephrase the sentence "There are too many authors, in order to improve efficiency and simplify the process of network construction, we finally choose high-yield authors to construct the rather than general authors."

page 6, section 2.3

- "and the matrix (Ii,j)K×K can also be represented as an upper triangular..." this is redundant.

- a cooperative relationship implies I_i,j != 0?

section 2.4

- when describing the network metrics please cite textbook material from network  / social network analysis

section 3

- why choose E=800?

- if you increase K and E the core authors remain the same?

- what do you mean by "when the values of both K and E were increased, the added nodes and edges increased the complexity of the network and couldn’t clearly show the structure of the it"?

- you repeat the same sentence a few times "The high-yield authors of the 466,118 papers during 2014-2018 were used.."

- the caption for figure 6 is on a different page than the figure

- please give more information in the captions of figures and tables

Author Response

(The authors gave the same response as above.)

Round 2

Reviewer 1 Report

The authors have addressed most of my concerns in this revised version of the paper. I still have some minor suggestions to make:

1. Authors replied to my last point that "Thanks to the reviewers’ suggestion, we have checked the researchers with the same name shown in Table 2". I think this verification should be mentioned in the paper to increase the credibility of the results.

2. Regarding my previous Point 9, I have not seen the inclusion of the values of gamma, the exponent of the power-law distributions shown in Figure 3 in Section 3.2. It would be good to see the evolution of this exponent along the years.

3. Page 10, lines 254-256. Authors state that: "Not only the number of nodes with higher degrees is increasing, but also the number of nodes with relatively low degrees is increasing, resulting in the increasing complexity of the network." However, from the graph in Figure 4, one cannot tell whether only the nodes with high degree centrality have increased the number of the connections or both high and low degree centrality nodes have increased their degree. On the other hand, it is important to highlight that although the mean degree centrality has a slight increase over the years, the median degree centrality remained stable, what implies that degree distribution is becoming more asymmetric over the years.

4. I advice the authors to make an English review of the entire paper since there are some typos and missing pronouns and articles.

Author Response

We have uploaded a point-by-point response  file to the reviewer’s comments

Reviewer 2 Report

The authors improved the paper quality. The paper can be accepted for publication.

Author Response

We have revised the paper and thank the reviewers for their thorough reviews, thoughtful comments and constructive suggestions.